# Serotonin-dependent kinetics of feeding bursts underlie a graded response to food availability in *C. elegans*

Kyung Suk Lee[1,2], Shachar Iwanir[3], Ronen B. Kopito[1], Monika Scholz[3], John A. Calarco[2], David Biron[3,4] & Erel Levine[1,2]

Animals integrate physiological and environmental signals to modulate their food uptake. The nematode *C. elegans*, whose food uptake consists of pumping bacteria from the environment into the gut, provides excellent opportunities for discovering principles of conserved regulatory mechanisms. Here we show that worms implement a graded feeding response to the concentration of environmental bacteria by modulating a commitment to bursts of fast pumping. Using long-term, high-resolution, longitudinal recordings of feeding dynamics under defined conditions, we find that the frequency and duration of pumping bursts increase and the duration of long pauses diminishes in environments richer in bacteria. The bioamine serotonin is required for food-dependent induction of bursts as well as for maintaining their high rate of pumping through two distinct mechanisms. We identify the differential roles of distinct families of serotonin receptors in this process and propose that regulation of bursts is a conserved mechanism of behaviour and motor control.

[1] Department of Physics, Harvard University, Cambridge, Massachusetts 02138, USA. [2] FAS Center for Systems Biology, Harvard University, Cambridge, Massachusetts 02138, USA. [3] Department of Physics and the James Franck Institute, The University of Chicago, Chicago, Illinois 60637, USA. [4] The Institute for Biophysical Dynamics, The University of Chicago, Chicago, Illinois 60637, USA. Correspondence and requests for materials should be addressed to E.L. (email: elevine@fas.harvard.edu).

The regulation of food uptake is a critical mechanism with major physiological impacts. Aberrant feeding behaviours undermine weight regulation and are associated with a range of chronic diseases, while dietary restriction has implications on health and longevity[1–6]. Simple model systems, such as the nematode *Caenorhabditis elegans*, have proven invaluable to understanding the mechanisms of regulation of food uptake[7,8]. Feeding of worms on bacteria is facilitated by the action of the pharynx, a neuromuscular pump that draws bacteria suspended in liquid into the gut from the surrounding environment and transports them to the intestine after concentrating and grinding[7,9–13]. This process is conducted by a combination of two feeding motions, pumping and isthmus peristalsis. Pharyngeal pumping is a cycle of contraction and relaxation of pharyngeal muscles, in which bacterial food is taken from the environment, bacterial cells are trapped and the surrounding liquid is expelled. Isthmus peristalsis is a peristaltic contraction of the muscles that carry the food within the pharynx[7,12]. Pharyngeal pumping is therefore a primary indicator of food intake.

The two feeding motions are operated by the pharyngeal nervous system, one of the two independently functioning nervous systems of the worm, consisting of 20 neurons of 14 types[14]. The MC and M3 neurons control pumping, while M4 is essential for isthmus peristalsis[10,11]. Previous results suggest that pharyngeal pumping depends on feeding history and quality of food through mechanisms that involve the neurotransmitter serotonin (5-hydroxytryptamine (5-HT))[15–18]. Serotonin increases feeding rate of *C. elegans*[19] and has been suggested as a putative food signal that controls feeding of the animal[20]. Evidence suggests that serotonin acts directly on MC and M3 neurons to facilitate fast pharyngeal muscle contraction–relaxation cycles[16,21,22].

Serotonin is a key modulator in the worm, involved in a variety of processes, including foraging, mating, egg-laying, metabolism, chemosensation, aversive olfactory learning and feeding[20,23–30]. Endogenous synthesis of serotonin requires the tryptophan hydroxylase, which is expressed in all serotonergic neurons from the *tph-1* gene[14,25,31]. Two pairs of head serotonergic neurons, the pharyngeal secretory neurons (NSM) and the chemosensory neurons (ADF)[18,20,32], have been implicated in feeding, although their precise roles remain elusive[32,33]. Additional complexity comes from the differential roles played by the different families of serotonin receptors. Three of the five known species of serotonin receptors, SER-1, SER-4 and SER-7, are expressed in pharyngeal neurons or muscles, suggesting a possible involvement in pharyngeal activity[34–36].

Here we hypothesize that worms modulate the dynamics of feeding in response to the availability of food in their environment and seek to characterize this response and trace its origins. Conventional feeding assays[37] are performed on dense bacterial lawns, which does not allow for fine control of food concentration. We therefore employed a custom microfluidic device[38] that enabled us to precisely control the concentration of available food and to monitor the dynamics of pharyngeal pumping at high resolution. Using these data, we show that feeding is characterized by bursts of fast, regular pumping, whose duration and frequency are correlated with the availability of food. In addition, feeding at low food densities is characterized by an abundance of long pauses. We implicate serotonin in promoting fast pumping and in suppressing long pauses and utilize these phenotypes to show that functional NSM neurons are necessary and sufficient for bursts of fast pumping and that serotonin biosynthesis in the NSM and HSN neurons is individually sufficient and together required for these bursts. This suggests that the HSN neurons, which are located in the middle of the worm body and send no processes into the pharynx, are involved in the regulation of feeding. We show that the 5-HT1 ortholog SER-1 is involved in food-dependent induction of fast pumping. Furthermore, our data suggest that the 5-HT2 ortholog SER-4 may be involved in maintaining the high pumping rate. Finally, we point out differences between induction of pumping by food and by exogenous serotonin, which relies on SER-4 and the 5-HT7 ortholog SER-7.

## Results

**Long-term automatic recording of pumping dynamics.** Typical measurements of pharyngeal behaviour involve manual scoring over short (30–60 s) intervals[37], to estimate the average number of pumping events observed per unit of time, or the average pumping rate. To allow more detailed characterization of feeding dynamics, we performed long-term time-lapse imaging of the pharynx of multiple worms (Fig. 1a and Supplementary Movie 1). For longitudinal measurements, worms were individually confined in a custom microfluidic device that permits maintenance of worms for >24 h with no detectable effect on their longevity, egg-laying or pumping dynamics (Supplementary Fig. 1)[38]. Bacteria suspended in standard S-medium were constantly flown through the device, ensuring a continuous supply of bacteria at a fixed density. In what follows, we indicate the densities of bacterial suspensions in units of optical density at 600 nm ($OD_{600}$).

In our experiments, worms were loaded to the device and fed high-density bacteria ($OD_{600} = 3$) for 3 h before transitioning to the test media and beginning measurements. We cycled through the worms, taking a 5–30 min time-lapse movie at a frequency of 50 frames per second for each worm in each imaging cycle. Our measurements lasted 2–3 h, allowing us to revisit most worms multiple times.

The resulting movies were analysed using a custom workflow that automatically detects pumping events by tracking the motion of the grinder. The high contrast between the grinder and its surroundings and its fast motion permit easy detection of pumping events by comparing consecutive images (Fig. 1b, Supplementary Movie 2 and Materials and methods). The intensity of the difference between images is indicative of the speed of the grinder motion (Fig. 1c), while the displacement between changing regions provides information about the direction of the motion (Fig. 1d). A pumping event is identified as a reset of the grinder (after the previous event), inversion of the grinder plates and relaxation of the muscle[12,13].

For each event, we define $\Delta t_{PD}$ as the time between contraction and relaxation in a single pumping event (pump duration (PD)), and $\Delta t_{PI}$ as the time interval between relaxation times of consecutive pumping events (pumping interval (PI); Fig. 1c,d). As each contraction–relaxation cycle corresponds to a single muscle action potential[39,40], the pulse duration $\Delta t_{PD}$ and the interpulse interval (IPI) $\Delta t_{IPI} = \Delta t_{PI} - \Delta t_{PD}$ are related to the duration of an action potential and the time the neuron membrane potential rebounds after repolarization, respectively.

**The average pumping rate increases with food concentration.** Food availability is presumed to be one of the key factors that regulate feeding. In worms, the presence of bacterial food on solid media stimulate pumping[20], and the mean feeding rate of worms grown in liquid culture increases with food concentrations[19], although worms grow slowly and show reduced fertility in liquid medium[41,42].

To characterize the regulation of feeding behaviour in response to food availability, we collected and analysed time-series data from multiple 'wild-type' worms (the laboratory strain N2)

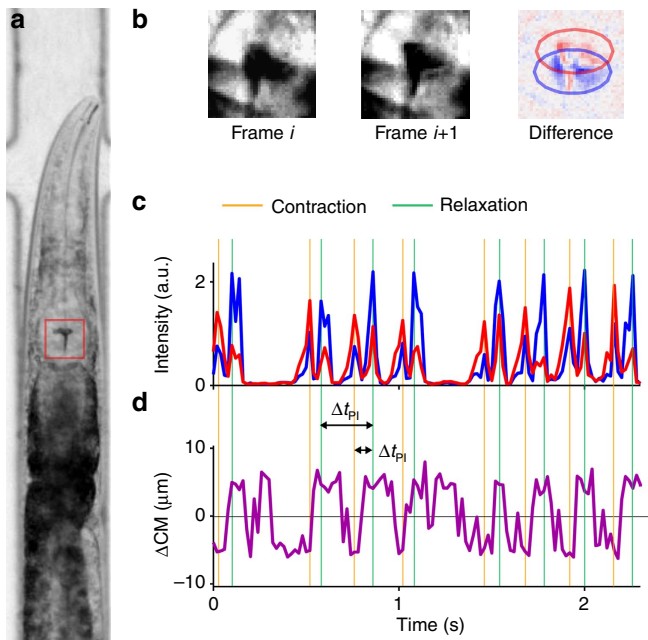

**Figure 1 | Automated detection of pumping dynamics.** (**a**) Bright-field image of a worm confined in a custom microfluidic device for longitudinal recording of pumping dynamics. (**b**) Images of a small region surrounding the pharyngeal grinder (red box in **a**) at two consecutive frames and the difference between the two. Pixels that turn darker and pixels that become brighter are coloured in red and blue, respectively. (**c**) Representative time courses of the total intensity of the red- and blue-marked pixels. Peaks in these measures are robust indicators of a pharyngeal motion, either contraction or relaxation. (**d**) Representative time courses of the displacement between the centrr of mass of the red- and blue-marked pixels. Positive values occur when the blue centre of mass is posterior to the red one, indicating a relaxation motion.

at different food concentrations in the range from $OD_{600} = 0$ to $OD_{600} = 8$. To connect with previous results, we first estimated the average pumping rate at each density by counting the number of pumping events observed in each worm, dividing this number by the duration of that measurement and averaging over all worms. We found that the average rate increases gradually with the concentration of supplied food from almost zero in clean media to a maximal average rate of 270 pumps per min in densities above $OD_{600} = 3$ (Fig. 2a).

The duration of individual pumps was unaffected by the density of food. To see this, we measured the relative frequency (that is, the empirical probability) of pump durations $\Delta t_{PD}$ at different food concentrations. We found that these distributions were always sharply peaked around 80 ms in all food concentrations, except in the complete absence of food (Fig. 2b and Supplementary Fig. 2A). Thus the duration of individual pumps is held fixed in fed worms and does not contribute to the regulation of food uptake. The modulation of feeding dynamics is therefore captured in the way pumping events occur in time.

**A bursting pumping dynamics.** An efficient way to represent the time series of pumping by an individual animal is to graph the cumulative count of pumping events it performs over time (Fig. 2c and Supplementary Fig. 2B). For each animal, the instantaneous pumping rate is the slope of a tangent line at any point on this graph, while the average pumping rate is the slope of a chord line of the same graph (Supplementary Fig. 2C). At high

food concentrations (above $OD_{600} = 3$), most worms pumped at a constant instantaneous rate, which was close to their average rate. In contrast, at lower food concentrations the pumping-count graphs alternated between periods of regular pumping at almost constant instantaneous rate (straight diagonal segments) and periods of much slower and irregular pumping (curved segments). In such cases, the average pumping rate fails to capture the true dynamics of feeding. This led us to hypothesize that pumping dynamics are not homogeneous in time. Rather, we propose that pumping occurs in bursts of fast, regular pumping punctuated with periods of slower, irregular pumping.

Support for the idea of inhomogeneous pumping dynamics comes from the relative frequency of different pumping intervals ($\Delta t_{PI}$) observed at different food concentrations (Fig. 2d). These histograms were typically composed of a sharp peak around 174 ms (corresponding to 6 pumps per second or 345 pumps per min) and a heavy tail of much longer intervals, extending up to several seconds. The short-interval peak, resembling a Gaussian distribution with an s.d. of 20 ms, dominated the histograms at high food concentrations, consistent with the observation that, at these densities, individual worms pump at a constant instantaneous rate (Fig. 2c and Supplementary Fig. 2B). Previous electrophysiological recordings from the pharynx demonstrate that a pumping action potential and the subsequent rebound after repolarization take around 250 ms[13], a timescale comparable to the pumping interval. This suggests that the pump intervals that contribute to the peak are as short as physiologically possible.

To help in classifying different pumping behaviours, Fig. 3a shows the same data as Fig. 2d in a different representation. For a given time $t$, we let $F(t)$ be the fraction of time spent by the population of worms in pumping intervals ($\Delta t_{PI}$) that are longer than $t$. For example, $F(600\,\text{ms}) = 10\%$ means that on average a worm at the relevant food concentration spends 10% of its time in pumping intervals that are $> 600\,\text{ms}$. In Fig. 3a, we plot the measured values of $F$ at different food densities as a function of the instantaneous rate $r = 1/t$, rather than $t$, as this representation clearly suggests 3 classes of pumping dynamics: long pauses (corresponding to the sharp decline at small $r$); fast pumping (or short intervals, the steep increase at large $r$); and intermediate intervals, for which—curiously—the curve shows steady linear increase with the rate $r$. This feature allows us to unambiguously define the fraction of time spent in long pauses and the fraction of time spent in bursts of fast pumping, respectively, as the value of $F(r)$ at the first point and the value $1 - F(r)$ at the last point of the linear segment in each curve in Fig. 3a. In almost all cases we studied, the former occurs around $r = 100$ pumps per min (corresponding to intervals $> 600\,\text{ms}$) and the latter around $r = 250$ pumps per min (or intervals $< 240\,\text{ms}$). For simplicity, we therefore refer hereafter to intervals $> 600\,\text{ms}$ as long pauses and to pumping at rate $> 250$ pumps per min as fast pumping, except when the range of the linear part of $F(r)$ is significantly different.

As exemplified in Fig. 2c, individual worms often display pumping dynamics of all three classes (Supplementary Fig. 3A). The hypothesis that fast pumping appears in bursts suggests that short intervals ($\Delta t_{PI}$) are clustered in time, rather than spread randomly among intermediate intervals and long pauses. To test this hypothesis, we compared the observed statistics of burst duration (red curve of Fig. 3b) with what is expected if the duration of different intervals were uncorrelated (black curve of Fig. 3b). In support of our hypothesis, we observed that the duration of bursts was significantly longer than expected by chance ($P$ value $< 10^{-50}$, Kolmogorov–Smirnov test).

**Modulation of fast pumping in response to food availability.** We next sought to characterize the effect food availability has on

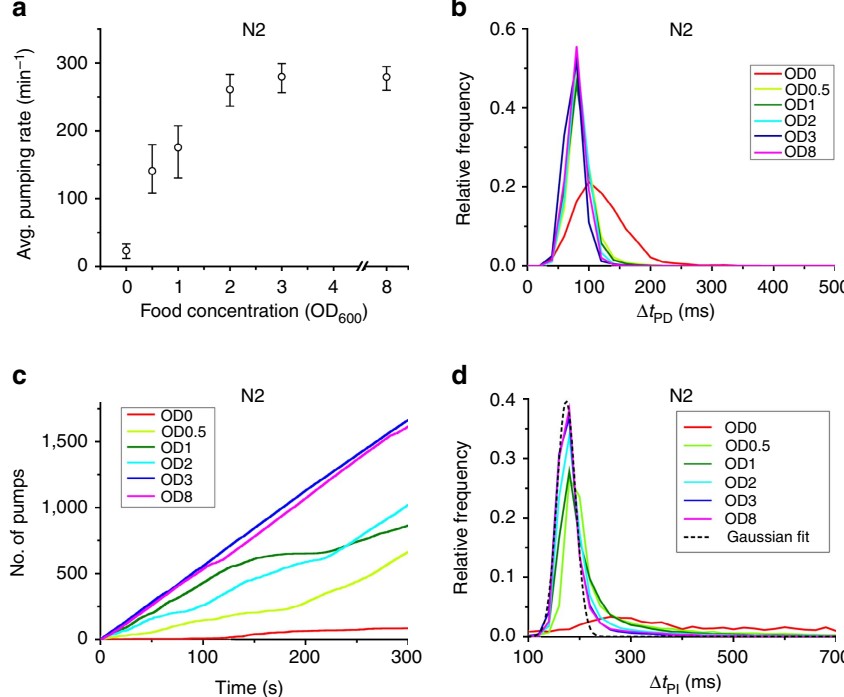

**Figure 2 | Pumping dynamics are modulated in response to food availability.** (**a**) Average pumping rate as a function of food concentration. The rates are averaged over all worms and all recorded periods at each concentration. Error bars are 95% confidence intervals, calculated from an ensemble of all 10-min subsets of the data. Sample sizes are given in Supplementary Table 2. (**b**) The distributions of pump duration $\Delta t_{PD}$. (**c**) Representative time series of pumping by individual worms at different food concentrations. Plotted is the total numbers of pumps counted since the beginning of the experiment (defined as time = 0). (**d**) Histogram of pulse intervals, $\Delta t_{PI}$. These histograms show a narrow peak and a heavy tail. One of these peaks was fit to a Gaussian distribution (black dashed line) centred at 174 ms, with the s.d. of 20 ms (adjusted $R^2 = 0.985$).

the dynamics of pumping. The data in Fig. 3a suggest that the density of food affects the fraction of time spent in both fast pumping and long pauses. In worms not supplied with food, pumping dynamics were dominated by long pauses. These long pauses also take up a significant fraction of time in worms fed with low-concentration food, but the fraction of time spent in such pauses decreases rapidly with the increasing density of food (Fig. 3c). Conversely, the time spent in fast pumping increases gradually with the concentration of food (Fig. 3d). The relative time spent in both types of dynamics saturates around $OD_{600} = 3$.

In the presence of food, most pumping events occur during periods of fast pumping (Fig. 2d). We therefore reasoned that the modulation of the average pumping rate in response to food comes mostly from modulation of the fraction of time spent in fast pumping. Indeed, we find a very strong correlation ($R^2 = 0.99$) between the average pumping rates at different food concentrations and the fraction of time spent in fast pumping (Fig. 3e). Variation in the fraction of time spent in bursts comes both from an increase in the average duration of individual bursts (Supplementary Fig. 3B) and from an increase in their frequency (that is, a decrease in the time intervals between them, Supplementary Fig. 3C).

**Serotonin production is required for fast pumping.** The neurotransmitter serotonin is known to stimulate pharyngeal pumping by affecting MC neurons to increase the frequency of pharyngeal pumping and by decreasing the duration of the action potential in M3 neurons[16,21,22,43]. In the absence of food, exogenous serotonin stimulates the activity of the MC neurons and induces pumping, mimicking the effect of food[20]. As our

results suggest that the relative frequencies of long pauses and bursts of fast pumping depend on food concentration, we asked whether serotonin is involved in the mechanism behind this relation.

To address this question, we first recorded the pumping dynamics in *tph-1* deletion mutants, which lack the tryptophan hydroxylase required for serotonin biosynthesis[25]. The fraction of time spent in fast pumping was significantly suppressed in these worms at all food concentrations (Fig. 4a). In fact, fast pumping was under-represented in these worms, as seen by the absence of a high-rate steep increase in Fig. 4b. As a result, the average pumping rate in *tph-1* worms was significantly lower than in wild-type worms at all food densities (Fig. 5a).

**The role of NSM and HSN in food-driven fast pumping.** Two pairs of serotonergic neurons reside in the head of the worm, the sensory neurons ADF and the pharyngeal neurons NSM. Another pair of serotonergic motor neurons, HSN, is located in the middle of the worm body and known to control vulval and uterine muscles. Although the ADF and NSM neurons were implicated in serotonergic modulation of feeding behaviour[18,20,32,44], their role remains elusive. In addition, the HSN neurons are not known to be involved in regulation of pumping.

To examine the role of these serotonergic neurons in modulating feeding dynamics in response to food availability, we collected time series of pumping from worms in which serotonin production was inhibited in subsets of these neurons. This was carried out using a Cre/Lox strategy, as previously described[30]. Briefly, a single copy of the *tph-1* gene and its native promoter flanked by Lox signals was introduced into the

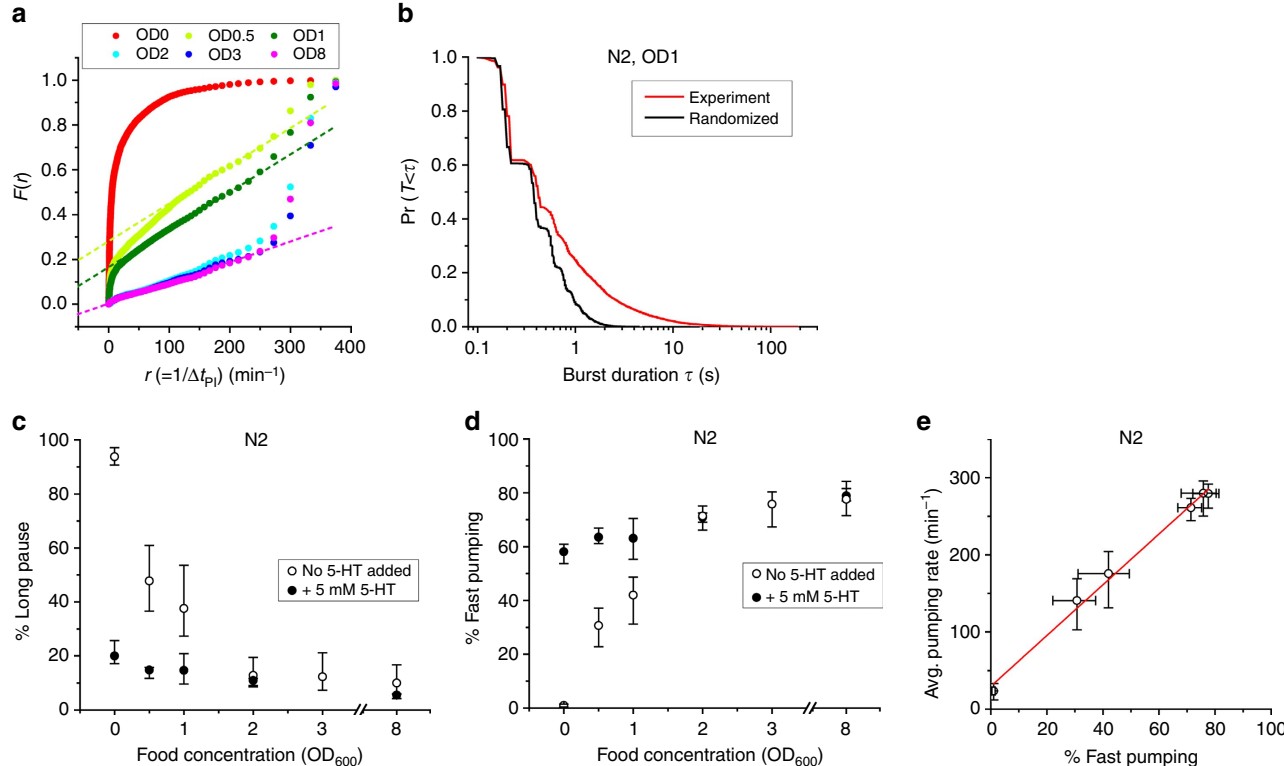

**Figure 3 | Bursts of fast pumping are a primary response to food availability.** (**a**) $F(r)$, the fraction of time spent in intervals longer than $1/r$ at various food concentrations. Dashed lines for ($OD_{600} = 0.5$, 1 and 8) are the best-fit lines between 100 pumps per min and 250 pumps per min. Sample sizes are given in Supplementary Table 2. (**b**) Survival probability of the durations of fast pumping bursts at $OD_{600} = 1$ (black). Red curve is the burst durations calculated from the same pumping intervals in randomized order. (**c,d**) The average fraction of time spent by worms in long pauses (**c**) and fast pumping (**d**) at different food concentrations. Empty symbols, no added serotonin; filled symbols, in media supplemented with 5 mM serotonin. Error bars as in Fig. 2. (**e**) Average pumping rates plotted against the fraction of fast pumping at different food concentrations. Red line is a linear fit to the data.

genome of *tph-1* worms. The *cre* gene was then introduced into these worms using neuron-specific promoters (either individually or in groups) to excise the *tph-1* gene only in those neurons. Inhibition of serotonin production in all three pairs of neurons (NSM, ADF and HSN) simultaneously led to strong suppression of fast pumping (Fig. 4c) and the elimination of the high-rate steep increase (Supplementary Fig. 4A) as observed in *tph-1* mutants (Fig. 4b), demonstrating the potency of the method.

In contrast, inhibition of serotonin production from each pair of neurons individually had a much less dramatic effect. Excision of the floxed *tph-1* either in the two ADF neurons or in the two HSN neurons had no noticeable effect on the dynamics of pumping. For ADF, this result is rather surprising, given the known role of ADF in regulation of feeding on agar surface[32] and for feeding response to familiar food[18]. We therefore confirmed this result using two independent approaches. First, we inhibited synaptic release of serotonin from the ADF neurons by expressing the tetanus toxin (TeTx) from the ADF-specific *srh-142* promoter[45]. These worms exhibited pumping dynamics indistinguishable from that of wild-type worms at both high and intermediate food concentrations (Fig. 4c and Supplementary Fig. 4A,B). Similarly, inhibition of the ADF neurons by transgenic expression of an activated form of potassium channel *twk-18(cn110)*[46] showed no significant effect on pumping dynamics. Together, these results demonstrate that, under these conditions, ADF neurons are not necessary for controlling the pumping dynamics, suggesting context-dependent functions of ADF, as discussed below.

Excision of floxed *tph-1* in the NSM neurons alone or in both the NSM and ADF neurons led to a small decrease in the fraction of time spent in fast pumping at high food concentration (Fig. 4c) but not at intermediate concentration (Supplementary Fig. 4B). In contrast, simultaneous inhibition of serotonin production in NSM and HSN neurons led to a substantial decrease in fast pumping to the same level observed in *tph-1* worms and in worms where floxed *tph-1* was excised from all three pairs of neurons. The remarkable difference between the case where *tph-1* was perturbed only in the NSM neurons and the cases where it was perturbed in both NSM and HSN suggests that the HSN neurons are involved in controlling pumping.

**NSM is necessary for food-driven fast pumping.** The HSN neurons are located away from the worm head and send processes to the nerve ring but not into the pharynx. We therefore hypothesized that, while HSN-produced serotonin can compensate for lack of serotonin biosynthesis in the NSM neurons, functional NSM neurons are nevertheless required for food-driven fast pumping. To test this hypothesis, we characterized the pumping dynamics in strains in which the NSM or HSN neurons were ablated by a reconstituted caspase[47]. In these strains, one of the subunits of caspase-3 was expressed from the *tph-1* promoter, such that it is only expressed in serotonergic neurons. The other subunit was expressed either from the *egl-6* promoter, which among the serotonergic neurons is only expressed in HSN, or from the *ceh-2* promoter, which

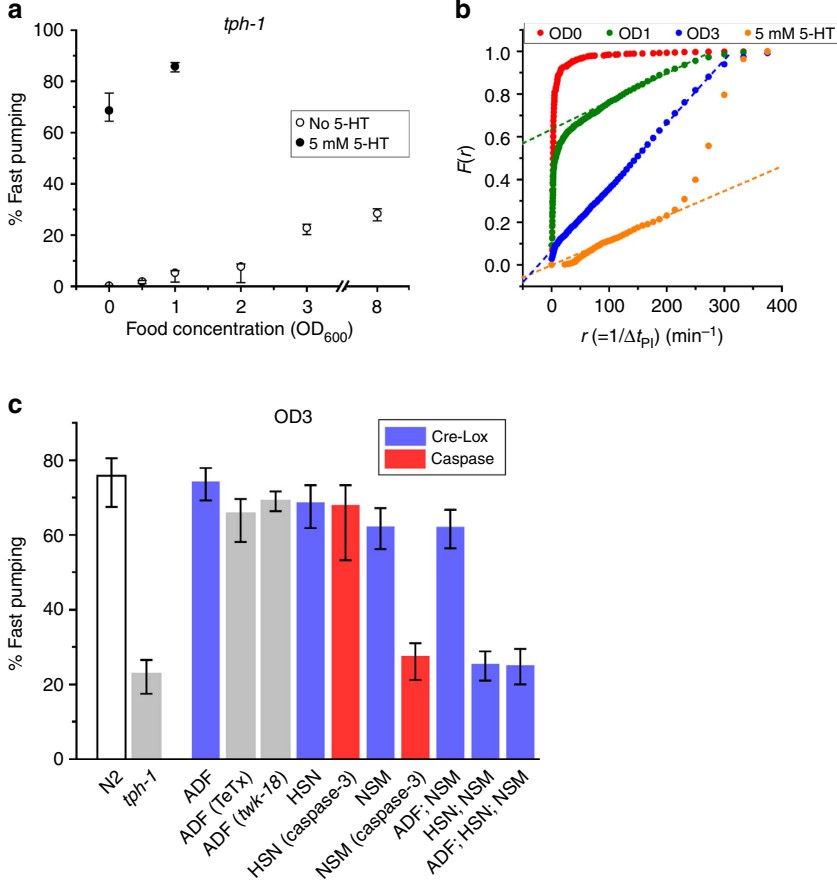

**Figure 4 | Serotonin is essential for induction of fast pumping.** (**a**) The average fraction of time spent in fast pumping by *tph-1* worms at various food concentrations. Empty symbols, no added serotonin; filled symbols, in media supplemented with 5 mM serotonin. Sample sizes are given in Supplementary Table 2. (**b**) $F(r)$, the fraction of time spent in intervals longer than $1/r$, in various food concentrations with no serotonin added to the media (red, green, blue) and in 5 mM serotonin in the absence of bacteria (orange). Dashed lines are the best-fit lines between 100 pumps per min and 250 pumps per min. (**c**) The average fraction of time spent in fast pumping by worms in which serotonin production or secretion has been perturbed in all or some of the serotonergic neurons (blue and grey) and in worms in which serotonergic neurons had been ablated by reconstituted caspase-3 (red). Food was supplied at $OD_{600} = 3$. Error bars as in Fig. 2.

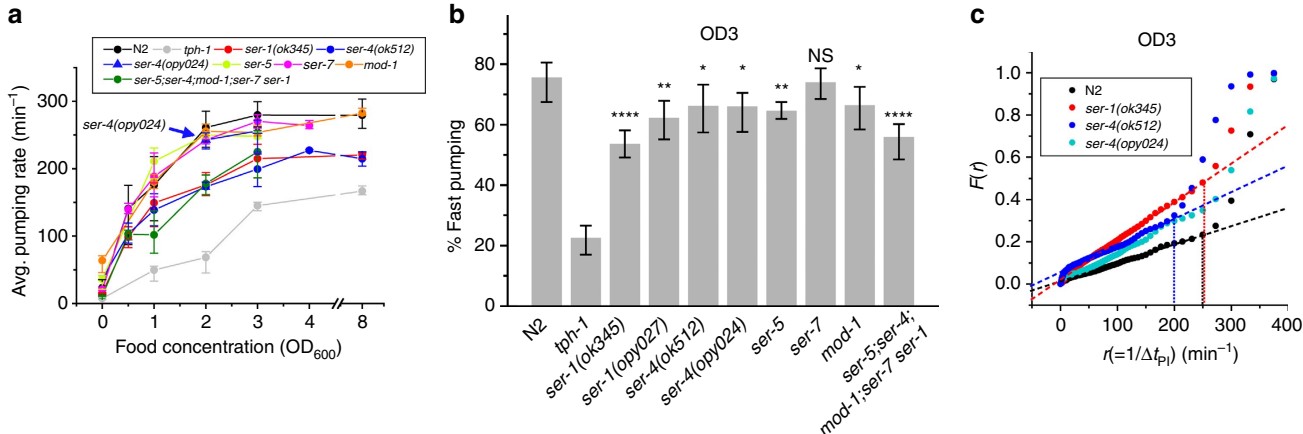

**Figure 5 | Distinct roles of serotonin receptors in food-driven pumping.** (**a**) Average pumping rates of wild-type (N2) worms, *tph-1* mutants and a range of serotonin receptor mutants at different food concentration. (**b**) The average fraction of time spent by these worms in fast pumping at $OD_{600} = 3$. Error bars as in Fig. 2, sample sizes in Supplementary Table 2. Here the threshold rate for fast pumping was taken to be 200 pumps per min for *ser-4(ok512)* worms and 250 pumps per min otherwise (based on **c** and on Supplementary Fig. 5A). The statistical significance of the difference from wild type is indicated (*$P < 0.05$; **$P < 0.01$; ****$P < 0.0001$; NS, $P > 0.05$). (**c**) $F(r)$, the fraction of time spent in intervals longer than $1/r$, for wild-type worms and *ser-1(ok345)*, *ser-4(ok512)* and *ser-4(opy024)* mutants at $OD_{600} = 3$, with the best fit lines (dashed) between 100 pumps per min and 250 pumps per min. The dotted vertical lines indicate the last point of the linear segment of $F(r)$, taken as the threshold rate for fast pumping, demonstrating its slowdown in *ser-4(ok512)* mutants.

among these neurons is only expressed in NSM. A simultaneous expression of tph-1p::GFP allowed us to verify directly the success and specificity of the neuron ablation (Supplementary Fig. 4C–F).

Animals in which the NSM neurons were successfully ablated did not exhibit bursts of fast pumping and an abundance of fast pumps that was similar to that of the tph-1 mutant. In contrast, ablation of the HSN neurons did not abolish the fast-pumping bursts and reduced the abundance of fast pumps only slightly (Fig. 4c, Supplementary Fig. 4G–I). These data suggest that the NSM neurons are required for normal pumping behaviour even when the HSN neurons are producing serotonin.

**SER-1 and SER-4 are involved in food-driven fast pumping.** Five serotonin receptors have been identified in C. elegans to date. Four are serotonin-activated G protein-coupled receptors (SER-1, -4, -5 and -7) and one is a serotonin-gated $Cl^-$ channel (MOD-1)[36,48]. The different receptors have individual patterns of expression and different (sometimes antagonistic) roles in serotonin-dependent behaviours. Thus the identities and functions of serotonin receptors involved in feeding behaviour is a complex question. Although the essential role of SER-7 in serotoninergic stimulation of pharyngeal pumping has received much attention[16,18,43], the excitatory roles of SER-1 (ref. 16) and SER-5 (ref. 32) as well as the inhibitory roles of SER-4 and MOD-1 (ref. 18) have also been discussed.

To investigate the roles of the different serotonin receptors in the modulation of fast pumping and long pauses in response to food, we considered worms carrying a null allele in each one of the known canonical serotonin receptors individually or in a quintuple mutant carrying all five simultaneously. Unlike tph-1 worms, which do not display bursts of fast pumping, all these strains showed consistent bursts in the presence of food. The average pumping rate, however, was consistently reduced at all food densities in worms carrying either one of two null alleles of two null alleles of ser-1 (ser-1(ok345) and ser-1(opy027)) and to a similar degree in worms carrying loss-of-function alleles of all five known genes that encode canonical serotonin receptors (Fig. 5a Supplementary Fig. 5A). A similar reduction has been observed in the fraction of time spent by these worms in fast pumping (Fig. 5b and Supplementary Fig. 5B).

Worms carrying a null allele of ser-4, which we created by removing seven out of eight exons of the gene using CRISPR-Cas9, showed consistent bursts of fast pumping, albeit at a slightly lower abundance than wild-type worms (Fig. 5 and Supplementary Fig. 5B). In contrast, worm carrying the ser-4(ok512) allele, a 1.3-kb in-frame deletion that removes transmembrane segment 5 and a part of the intracellular loop that connects it to transmembrane segment 6 (ref. 49), demonstrated a significant reduction in the maximal attainable pumping rate. In these worms, bursts of fast pumping were characterized by pumping that occurs at a rate of 200 pumps per min (cf. 250 pumps per min in wild-type and ser-4(opy024) worms). This was again determined by plotting $F(r)$, the fraction of time spent in different pumping intervals (Fig. 5c), and identifying the fast rate as the rate of the final ascent. The reduction in the fast rate came from prolonged intervals between pumping events ($\Delta t_{PI}$) and not from change in the duration of individual events ($\Delta t_{PD}$) (Supplementary Fig. 5C).

Mutations in the two other serotonin receptors, ser-5 and mod-1, resulted in weak suppression on fast pumping, while mutation in ser-7 showed no effect on pumping dynamics on food, as observed before[16] (Fig. 5b and Supplementary Fig. 5B,D).

**Exogenous supply of serotonin promotes fast pumping.** On agar plates, high concentration of serotonin activates overall feeding[25,43]. To link this behaviour to food-driven feeding, we recorded the dynamics of pumping in unfed wild-type worms and serotonin receptor mutants supplied with serotonin at different concentrations (Fig. 6a). Mimicking the effect of food, higher concentrations of serotonin in the media led to higher average pumping rate. Also, similar to food, increased concentrations of serotonin resulted in suppression of long pauses (Fig. 6b) and in an increase of the fraction of time spent in fast pumping (Fig. 6c). However, a clear difference was observed in the distribution of long pauses. In the presence of exogenous serotonin at concentrations $\geq 5$ mM, very long pauses, lasting several seconds, were effectively eliminated (Supplementary Fig. 6A,B,G). Such long pauses were interrupted by a few pumping events, even when these did not form a significant burst (Supplementary Fig. 6C). The dual effect of exogenous serotonin persists even in the presence of food (Fig. 3c,d). In addition, supplying exogenous serotonin allowed tph-1 worms to exhibit fast pumping with dynamics similar to that of wild-type worms (Fig. 4a,b)[43].

Induction of feeding by serotonin is known to depend on the serotonin receptor SER-7 (refs 16,43). Indeed, even in the presence of high concentrations of serotonin (5 mM), ser-7 worms spent a considerable fraction of their time in long pauses (Fig. 6d) and displayed no fast pumping (Fig. 6e). A similar but weaker phenotype was observed in ser-4 worms (carrying either the ser-4(ok512) allele or the CRISPR deletion ser-4(opy024)) for both effects. As in food-driven pumping, the rate of fast pumping by ser-4(ok512) animals was slower than that of wild-type worms (Supplementary Fig. 6H). Worms lacking all five canonical receptors showed the strongest phenotype, showing almost no response to exogenous serotonin and spending most of their time in long pauses.

## Discussion

The energy needs of a living organism must be fulfilled and balanced despite large fluctuations in nutrient availability and energy expenditure. Energy uptake is therefore regulated in response to the internal state of the body and to environmental conditions. Simple model organisms, such as C. elegans, are particularly useful for elucidating underlying regulatory mechanisms. The advance in microfluidics and image processing techniques offers a unique opportunity for a longitudinal study of the regulatory response of living animals to precisely defined environments.

In this work, we showed that the rate of pharyngeal pumping in the worm is continuously matched with the concentration of bacterial food in the environment. We found that the time series of pumping in individual worms occurs in bursts of fast pumping. During these bursts, worms display fast, regular pumping, at a rate similar to the maximal rate allowed by the underlying neuronal physiology[13]. The response in average feeding rate to food availability is controlled by modulating the fraction of time spent in fast pumping, rather than by continuously tuning the instantaneous pumping rate. Slow and irregular pumping has been previously described and linked to the effect of mutations that affect the MC neurons or the effect of starvation[11,13,21,50]. Our data suggest, however, that slow pumping also occurs in healthy animals even in the presence of food as part of regulation of the overall feeding rate.

The biogenic amine serotonin (5-HT) has a pivotal role in energy homeostasis, for example, by coordinating the perception of nutrient availability and feeding. Worms that cannot synthesize serotonin are capable of fast pumping but fail to generate bursts. This result is consistent with the seminal work of Hobson et al.[16], which showed how the worms fail to maintain

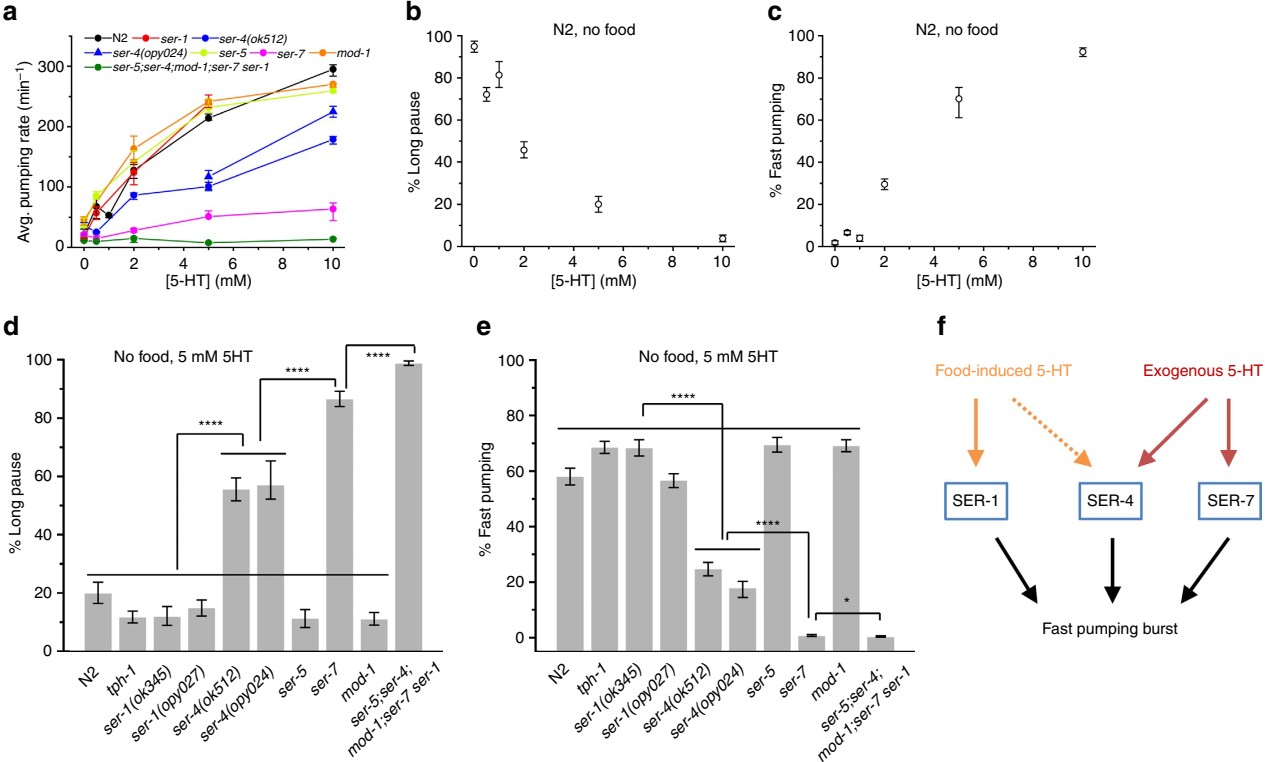

**Figure 6 | Distinct roles of serotonin receptors in pumping stimulated by exogenous serotonin.** (**a**) Average pumping rate of wild-type (N2) worms, *tph-1* mutants and a range of serotonin receptor mutants at different concentrations of serotonin (5-HT) in bacteria-free media. Sample sizes are given in Supplementary Table 2. (**b,c**) The average fraction of time spent in long pauses (**b**) and fast pumping (**c**) by wild-type (N2) worms as a function of the concentration of serotonin in bacteria-free media. (**d,e**) The average fraction of time spent in long pauses (**d**) and fast pumping (**e**) by wild-type (N2) worms, *tph-1* mutants and a range of serotonin receptor mutants in bacteria-free media supplemented with 5 mM serotonin. Here the threshold rate for fast pumping was taken to be 160 pumps per min for *ser-4* worms and 200 pumps per min otherwise (based on Supplementary Fig. 6A,F). Error bars as in Fig. 2. Statistical significance indicated as in Fig. 5. (**f**) Summary of the distinct roles of serotonin receptors. In food-driven pumping, SER-1 is involved in promoting pumping bursts, presumably with contributions from SER-4. The stimulation of pumping by exogenous serotonin requires the SER-7 and SER-4 receptors.

fast pumping over time when feeding on lawns of bacteria on agar plates.

We demonstrated that serotonin production in the pharyngeal motor neurons NSM and in the mid-body motor neuron HSN is necessary and sufficient for fast pumping in response to food. The involvement of NSM is consistent with previous reports that implicate the serotonin-controlled MC neurons in regular pumping[11,13,16] and complements studies of feeding under high food concentration conditions[32,33]. On the other hand, the role of HSN in the regulation of feeding is unexpected owing to their distal location from the pharynx and their known roles[51]. Recent findings, however, demonstrate a synergistic role for HSN and NSM in promoting a locomotion behavioural state known as dwelling, where worms inside a bacteria lawn move slowly and turn frequently to explore locally. This behaviour relies on serotonin production in both the NSM and HSN neurons but not in ADF[26,52].

Although functional NSM neurons are required for bursts of fast pumping and for enrichment of fast pumping in the presence of food, serotonin production in HSN is sufficient to elicit food-dependent pumping dynamics when serotonin production in NSM is perturbed. This suggests two possible models. First, it is possible that serotonin produced by HSN is being taken up by the NSM neurons (perhaps through the reuptake receptor MOD-5, which is expressed in the NSM neurons) and is used by NSM to transmit food-dependent

information to its postsynaptic cells. In this model, serotonin from HSN can be used by the NSM as an alternative source of this neurotransmitter in the absence of its *tph-1*. An alternative model suggests that humoral serotonin from HSN primes pharyngeal neurons to respond to serotonin-independent signaling from the NSM neurons.

It should be remarked that previous studies reported that worms whose NSM neurons were laser ablated displayed 'normal pumping'[11,18], taken to mean that these animals' pumping rate fell within 200–300 pumps per min[11] or rates that averaged at 200 pumps per min[18]. These results are consistent with our data, as we observe that bursts of fast pumping occur at rates faster than 260 pumps per min, and indeed even at low food densities when fast pumping is suppressed worms spend >40% of the time pumping at rates >200 pumps per min (Fig. 4b and Supplementary Fig. 4G). To observe that NSM ablated worms do not exhibit wild type-like behaviour require a more stringent definition.

The serotonergic sensory neurons ADF have been previously implicated in pumping on plates and regulation of feeding in response to familiar food[18,32]. It was therefore surprising that ADF had no role in modulating pumping dynamics in our experiments, and we verified these results using three independent approaches: excision of *tph-1* exclusively in the ADF neurons, inhibition of synaptic release of serotonin from ADF by expressing the TeTx, and inhibition of the ADF neurons

by transgenic expression of an activated form of potassium channel TWK-18. In addition, previous studies of the roles played by serotonergic neurons in food-associated stimulation of pharyngeal pumping led to contradicting results supporting either ADF or NSM[32,33].

Previous studies have linked ADF with stress response[28,53,54] and feeding in response to change of food quality[18]. The role of ADF in feeding regulation has been demonstrated for worms on a plate, where worms move freely in and out of the bacteria-covered region[32]. In contrast, in our experiments, well-fed worms were assayed in stationary environments, where the animals are kept at constant food concentration for hours. Thus it is possible that the roles played by the sensory neurons ADF are related with sensing fluctuations in the environment[18] that are suppressed in our experiments. ADF-dependent pumping regulation may also coincide with their role in regulating locomotion during encounter with food[52]. Consistently, we did not observe a clear phenotype in the mutant lacking ser-5, which has been associated with ADF-dependent feeding[32]. As an additional example of a difference between the two types of environment, the average pumping rates of tph-1 mutants were 28% and 52% of the wild-type rates at $OD_{600} = 1$ and 3, respectively, while a previous study of worms on plates[32] found it to be 80% of the wild-type rate (Supplementary Fig. 5G).

Previous anecdotal evidence raise the possibility that serotonin is only synthesized in the HSN neurons of adult worms but not in larvae[25,55]. This raises the possibility that in larvae fast-rate pumping requires serotonin production in the NSM neurons. This, however, turns out not to be the case, as fast pumping is abundant in wild-type L4 larvae as well as in animals in which tph-1 is excised in NSM neurons (Supplementary Fig. 4J). Although this result may indicate that the requirement for serotonin is different in larvae and in adults, it may also indicate that the low levels of serotonin in HSN, which were undetectable in previous experiments, are sufficient to support fast pumping in the larva[56].

Five serotonin receptors have been previously identified in the worm. These receptors exhibit differential expression patterns and have distinct, sometimes antagonistic roles in different behaviours. With this in mind, we characterized the roles of these receptors in modulation of long pauses and fast pumping in response to food. ser-1 worms spent significantly less time in fast pumping than wild-type worms and pumped less on average at all food concentrations. Mutations in other receptors, including ser-4, ser-5 and mod-1, led to a small decrease in fast pumping. With large sample size, these changes may be statistically significant. However, it is not clear whether this implies a functional role for these receptors in the wild-type animal.

Two alleles of ser-4 genes yield surprising results. A complete deletion of the gene using CRISPR-Cas9 showed only minor effect on the fast pumping dynamics in the presence of food. Worms carrying the previously characterized partial-deletion allele ser-4(ok512) demonstrated a similar weak phenotype in the abundance of fast pumping, yet the rate of fast pumping was slower than wild-type worms. As this allele had no effect on the duration of individual pumps (Supplementary Fig. 5C), ser-4(ok512) is likely to display abnormal contraction, rather than relaxation, of pharyngeal muscles. This implies a negative effect of the ser-4(ok512) allele. Thus our results suggest that SER-1 receptors are involved in promoting bursts of fast pumping, while SER-4 receptors could be involved in establishing their fast rate. In further support of a role for SER-4 in regulation of feeding, food-driven pumping is more suppressed in ser-7;

ser-4(opy024) double mutants than in the ser-7 worms (Supplementary Fig. 5E,F).

Activation of pumping by exogenous serotonin is markedly different from the dynamics induced by food. In the presence of 5 mM serotonin, we could not observe long pauses that extend beyond a few seconds (Supplementary Fig. 6G), while such pauses and longer were observed in feeding worms in all food concentration (Supplementary Fig. 6B). An increased concentration of food induces an increase in average pumping rate and in fast pumping, with complete correlation between the two (Fig. 3e). In contrast, while serotonin induces both fast pumping and an increase in the average pumping rate, the correlation between the two is nonlinear (Supplementary Fig. 6D). Also the rate of fast pumping induced by exogenous serotonin was slightly lower than that of food-induced pumping (Supplementary Fig. 6A). Finally, as noted before, serotonin-induced activation of pumping requires the SER-4 and SER-7 receptors, while absence of the latter has no significant effect on food-driven pumping. Together, our results suggest a model, in which food-driven fast pumping involves the SER-1 receptors presumably with contributions from the SER-4 receptors, while stimulated pumping by exogenous serotonin involves the SER-4 and SER-7 receptors (Fig. 6f). Interestingly, previous findings show that feeding on bacterial lawns on agar plates ser-1 and ser-7 mutants pump at an average rate similar to that of wild-type animals, but that their pumping is somewhat more irregular, while ser-7 ser-1 double mutants fail to maintain fast pumping and display reduced pumping rate[16]. In our hands, no significant difference was detected between the fraction of time spent in fast pumping in ser-1(ok345) and the ser-7 ser-1(ok345) double mutant.

Coupling between microfluidics and automated imaging enabled us to collect long-term longitudinal data from a large number of worms at high temporal frequency and in precisely defined environments. With these data, we were able to identify the multifaceted nature of the pumping dynamics, quantitatively link attributes of the environment and of the animal's behaviour and characterize the distinct roles of serotonergic neurons and serotonin receptors. We demonstrate that modulation of fast pumping bursts is a primary mechanism of feeding regulation to food availability in the environment. Although the fact that serotonin has role in controlling pumping is known for a long time, analysis of long traces of pumping reveals details of these roles and of those played by serotonergic neurons. In particular, we show that serotonin is essential for bursts of fast pumping. These bursts require the NSM neurons, but these neurons do not have to be the source of serotonin, as serotonin produced in HSNs is sufficient for the presence of bursts. It is interesting to ask how this cooperative function of the two distal neurons is implemented.

Feeding in bouts has been observed in other species. In rodents, feeding is organized by bursts of licking, and the increase in feeding following starvation is mainly due to shortening of the interval between bursts (the OFF mode)[57,58]. A recent study based on automated imaging of feeding behaviour in Drosophila demonstrated a similar behaviour in flies[59]. Increase in feeding after short-term starvation was achieved by a decrease in the interval between bursts, while a long-term starvation resulted in an increase of the burst length. In addition, the role of serotonin as a neuromodulator of feeding is conserved from invertebrates to mammals. The approach and tools described in this work, along with the relative simplicity of the worm's neuronal circuitry and anatomy, open the way to further investigate the dynamics of feeding and the coupling between physiological and molecular processes, neuronal circuitry and metabolism.

## Methods

**Strains.** All strains were maintained on standard nematode growth medium plates seeded with *Escherichia coli* strain OP50 (ref. 60) at 15 °C. Adult hermaphrodites were used in all assays. The N2 Bristol strain was used as wild-type animals, in addition to the following strains: MT15434 *tph-1(mg280)II*, INV33006 *Ex [tph-1p::TeTx::mcherry unc-122p::gfp]*, INV30001 *Ex [srh-142p::TeTx::mcherry unc-122p::gfp]*, ZC1890 *mgIS71V; yxEx960 [srh-142p::twk-18(cn110)::mcherry; unc-122p::dsred]*[46], CX13571 *tph-1(mg280)II; kySi56 IV; kyEx4077[srh-142p::nCre]*, CX13572 *tph-1(mg280)II; kySi56 IV; kyEx4057[ceh-2p::nCre]*, CX13576 *tph-1(mg280)II; kySi56 IV; kyEx4107[egl-6p::nCre]* CX15658 *tph-1(mg280)II; kySI56 IV; kyEx5262 [ceh-2p::nCre, egl-6p::nCre]*[30], DA1814 *ser-1(ok345)X*, AQ866 *ser-4(ok512)III*, RB2277 *ser-5(ok3087)I*, DA2100 *ser-7(tm1325)X*, DA2109 *ser-7(tm1325) ser-1(ok345) X*, and RWK3 *ser-5;ser-4(ok512);mod-1;ser-7(tm1425) ser-1(ok345)*[48]. Also the following strains were created: ERL76 *tph-1(mg280)II; kySi56 IV; opyEx18 [ceh-2p::nCre, srh-142p::nCre, myo-3::mCherry]*, ERL82 *tph-1(mg280)II; kySi56 IV; opyEx19 [ceh-2p::nCre, srh-142p::nCre, egl-6p::nCre, myo-3::mCherry]*, ERL96 *ser-7(tm1325); ser-4(opy024[myo-2p::gfp])*, ERL100 *mgIs71 [tph-1p::gfp, rol-6(su1006)] V; opyEx21 [ceh-2p::cz::caspase-3(p17); tph-1p::caspase-3(p12)::nz; myo-3:: mCherry]*, ERL104 *mgIs71 [tph-1p::gfp, rol-6(su1006)] V; opyEx22 [egl-6p::cz::caspase-3(p17); tph-1p::caspase-3(p12)::nz; myo-3p::mCherry]*, ERL105 *ser-1(ok345)X; opyEx23 [ser-1(+); rol-6; myo-3p::mCherry]*, ERL107 *ser-1(opy027[myo-2p::gfp])*, and ERL109 *ser-4(opy024[myo-2p::gfp])*.

**Creation of transgenic rescue strains.** To construct a *ser-1* transgenic rescue strain, the entire *ser-1* gene from the promoter to the 3′ untranslated region was PCR amplified from the N2 genome. PCR products were directly injected to *ser-1(ok345)* worms with co-injection markers (pRF, pCFJ104). The concentration of the PCR product was reduced to $0.05\,ng\,\mu l^{-1}$ in order to acquire a stable line. All primers are found in Supplementary Table 1.

**Creation of CRISPR alleles of serotonin receptor mutants.** CRISPR-Cas9 was used to excise the *ser-1* or *ser-4* genes, following the protocol of Norris et al.[61]. Briefly, sgRNA sequences were synthesized *de novo* and ligated into the *klp-12* sgRNA plasmid. To make homology repair vectors, homology arms were PCR amplified from the N2 genome and assembled into the *myo-2::GFP* neoR loxP disruption/deletion vector by Gibson assembly. All constructs were verified by sequencing. For each gene deletion, two sgRNA vectors and the homology repair vector were injected to worms, along with three co-injection markers. Successful integration was verified by selection on Neomycin, loss of the transient co-injection markers, stable inheritance of the integrated *myo-2::GFP* marker and by PCR amplification of the expected homology arms. Two independent lines were characterized for each gene. All primers are found in Supplementary Table 1.

**Imaging pharyngeal pumping dynamics in a microfluidic device.** For feeding experiments, an overnight culture of *E. coli* OP50 grown in LB media was centrifuged and resuspended at the required concentration in S-medium[62]. The bacterial solution was filtered through a 5 µm syringe filter to remove big aggregates, which could block the flow in the microfluidic device.

Age-synchronized worms were obtained by letting 15–20 gravid adults, incubated at 25 °C for at least a day, lay eggs on a plate for an hour. Adult worms were then removed, and the plates were incubated at 25 °C for 56–60 h. Worms were mixed into a bacterial solution and loaded into the microfluidic devices (Supplementary Fig. 1) as described previously[38].

Bacterial solution was injected into the device at the rate of $10\,\mu l\,min^{-1}$. To ensure that bacteria do not clog the device, we applied a 1-min pulse of $150\,\mu l\,min^{-1}$ up to three times per hour. Data acquisition started 3 h after feeding the animals at constant food concentration of interest to establish a baseline pumping dynamics. During the 3 h, individual animals were monitored longitudinally every 30 min–1 h, and animals that did not show normal head movement or did not show normal fast pumping at any time were discarded. After censoring, 10–15 worms were measured multiple times for each experiment. Each measurement of each animal lasted ≥ 5 min. For the titration experiments (Fig. 5a,b) of strains other than N2, the same set of worms was monitored at multiple conditions, as we decrease the concentrations of food and exogenous serotonin, respectively, step by step, with 30-min wait time at each step. All experiments were carried out in at least two independent biological repeats; experiments that showed subtle or unexpected phenotypes were carried out in up to six independent repeats. CRISPR-based experiments were carried out using two independent lines for each phenotype.

Imaging was carried out using a Zeiss Axio Observer.Z1 microscope with N-Achroplan 10 × objective (NA 0.25), a Hamamatsu ORCA Flash4.0 camera and ZEN 2012 software from Zeiss. Every worm was imaged for 5 min at 50 frames per second at spatial resolution of 0.65 µm per pixel. When required, representative worms were recorded for 30 min.

**Image analysis.** Extraction of pumping trajectories from raw images was carried out using custom software written in MATLAB. To analyse time-lapse images, the position of the grinder was identified in each frame. The difference in the brightness of pixels between every pair of consecutive frames was computed to detect contraction of pharyngeal muscles and subsequent relaxation (Supplementary Movie 2). We identified the centre of mass (CM) of pixels that become darker (marked in red in Fig. 1b,c) and the one of pixels that become brighter (blue). We then calculated the total difference in pixel brightness in an ellipse of constant area around each (Fig. 1c). Peaks in this measure are clear indicators of a rapid change in the pharynx position, either contraction or relaxation. To differentiate the two, we used the fact that the CM of the red pixels was posterior ($\Delta CM < 0$) to that of the blue pixels during contraction and anterior ($\Delta CM > 0$) to it during relaxation (Fig. 1d). Comparison of the total difference in brightness with the image background provides a confidence measure. Low confidence events were inspected manually and were either validated or discarded from further analysis. All scripts will be shared by the authors upon request.

**Statistical analysis.** For error bars, 95% confidence intervals were calculated from an ensemble of all 10-min samples from the data. To assign statistical significance to the difference between two conditions, we generated an empirical distribution of the differences between the 10-min samples in the two data sets and tested the hypothesis that the mean of this distribution is different than zero using Z-test.

**Data availability.** All relevant data are available from the authors upon request.

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

## Acknowledgements

We thank Carlos Riberio and Yun Zhang for discussions, Richard Komuniecki for reagents and technical advice and Mark Alkema, Yun Zhang and Cori Bargmann for reagents. This research was supported by the National Science Foundation through grants PHY-1205494, MCB-1413134 (to E.L.) and IOS-1256989 (to D.B.).

## Author contributions

K.S.L., D.B. and E.L. conceptualized this research. K.S.L. and E.L. designed the experiments, analysed the data and wrote the paper. K.S.L. developed the assay and performed the experiments. R.B.K. designed the microfluidic device. S.I., J.A.C. and D.B. provided reagents. K.S.L., M.S., D.B. and E.L. participated in discussions.

## Additional information

**Competing financial interests:** The authors declare no competing financial interests.

