## [Peer Review File · Nature Communications]

Reviewers' comments:

Reviewer #1 (Remarks to the Author):

Overall, this is an interesting study. The authors used a custom microfluidic device to deliver precise concentrations of bacterial food to worms while monitoring pharyngeal pumping. The use of this method to monitor pharyngeal pumping is highly innovative and should become the standard in this particular area of research. Increasing food concentration increases the duration and frequency of fast pumping bursts, but each pumping action is of uniform duration regardless of food concentration. The authors use a variety of mutants to demonstrate that serotonin underlies this behavior. The serotonergic neurons NSM and HSN are implicated in for fast pumping behaviors. The receptors SER-1 and SER-4 mediate two different aspects of this behavior.

Major:

The major criticism is that the article does not reveal a profoundly novel insight. The authors admit that serotonin is already known to mediate pumping behaviors. The article does contribute many novel findings to the field, including a role for the HSN neurons in regulating pharyngeal behavior and the specific roles of SER-1 and SER-4 receptors in mediating this behavior. The method developed by the authors is highly impressive, and is poised to become the gold standard for this type of research. However, the article would be strengthened in several ways by combining their approach with more sophisticated worm genetics. For instance, the authors use tetanus toxin expression and *twk-18(cn110)* expression in ADF to disrupt ADF signaling and activity, but they don't extend this approach to HSN or NSM neurons to confirm their requirement for pharyngeal pumping. In no case do they kill any of these neurons, either genetically or with laser ablation. This is especially warranted in the case of HSN, which appears to play a novel role in regulating pharyngeal pumping. They suggest the possibility that HSN may be secreting humoral serotonin, which would bypass NSM signaling. Killing NSM in their experiment would test this possibility directly.

The authors used five null mutants for five serotonin receptors, and one quintuple mutant. While impressive, the use of double mutants would allow for more sophisticated analyses. For instance, the authors cite that *ser-7;ser-4* double mutants have been shown to exhibit behavioral phenotypes not seen in single mutants on agar plates. How would that same double mutant behave in their experimental design? If there are differences between worms on plates and worms in microfluidic plates (the authors highlight several), then direct comparisons of mutant behavior between them are inappropriate.

Other comments:

Given that one of major features of this article is their combination of microfluidics with high resolution imaging, it would be good if they could identify the software used for imaging. If it is custom software, then publishing the code would be helpful.

In Figure S2, the x-axis label is cut off.

In Figure S2C, the figure indicates the solid lines are the average of the dotted line values. I believe the solid lines are actually the chord lines. If this is the case, then the figure should indicate that.

I was unable to view the movies, so I have no comments on those.

Reviewer #2 (Remarks to the Author):

The manuscript by Lee et al. describes a method monitoring *C. elegans* feeding behavior in liquid media. The authors made use of a microfluidic device, which is composed of an array of chambers with each chamber housing one worm, and performed time-lapse imaging of pharyngeal pumping of individual worms. This group has already published the design and utility of the device. In this manuscript, the authors characterized pumping frequency and duration of individual pumps as a function of bacterial food concentration in the media. Further, the authors tested a series of mutants in neurotransmitter serotonin synthesis and serotonin receptors, and showed that multiple serotonergic neurons and serotonin receptors were involved in food-dependent increase of pumping rates. Although many methods have been used in characterizing *C. elegans* feeding behavior, this method appears easy to use, especially for quantifying detailed pumping events. Likewise, serotonin synthesis and serotonin receptor mutants in food-induced pumping have been reported by several laboratories, but this manuscript identified the involvement of another pair of serotonergic neurons, NSM, in modulating pumping behavior. Overall the work is interesting although it is not entirely new.

The following are specific comments to the experiments and data:

1. The role of serotonin in the current feeding assay needs to be clarified. The authors showed that the ADF serotonergic neurons, which have been shown to play a key role in feeding behavior by several laboratories, are not required for the pumping events measured in this assay. In contrast, the authors identified that a pair of HSN serotonergic neurons plays an important role in the current assay. Since HSN neurons do not synthesize serotonin in *C. elegans* larvae, it would be interesting to determine if serotonin regulates feeding behavior only in adult but not in larval worms. This is a simple experiment, should be done.

The authors suggested that one possibility is that NSM neurons uptake serotonin released from HSN to in turn regulate feeding. This possibility is unlikely, as laser ablation of NSM did not reduce pumping rates (Avery & Horvitz, Neuron, 1989). In fact, biochemical pathways and neural circuits involved in *C. elegans* feeding behavior have been studied extensively and many mutants are available. Some experiments to place serotonin and serotonergic neurons into the behavioral circuits could offer new mechanistic insights into serotonin regulation of feeding behavior, and enhance general interests to the paper.

2. Experiments over serotonin receptor mutants need to be improved to make the points clear. Figure 5B showed that serotonin synthesis mutant *tph-1* was defective in fast pumping, but deletion mutants of any of the five identified *C. elegans* serotonin receptors or quintuplet mutants of all the serotonin receptors largely preserved fast pumping frequency, suggesting that serotonin can stimulate pumping in the absence of these receptors. However, Figure 6E showed that the quintuplet serotonin receptor mutants completely failed to respond to exogenous serotonin. One caveat is that the data were entirely rested on

single mutant allele of the receptors. The authors should validate the results by testing multiple available serotonin receptor alleles, and performing transgenic rescue of the phenotype. In addition, to determine gene function in behavior, typically sibling worms of each genotype with or without environmental stimuli or drug treatments should be assayed in parallel, thus the basal pumping rates and the ability to respond to food and drugs be compared.

3. The authors should provide more details about the experiments: how many animals were tested for each experiment, how many independent trials were performed for each assay, and which differences were statistically significant.

Reviewer #3 (Remarks to the Author):

Summary:

Lee et al developed a new method to measure how feeding motion is regulated by food availability and concentrations in *C. elegans* using microfluidic device and long-term imaging tools. From the study, they discovered that *C. elegans* regulates food intake by altering the frequency of bursts of fast pumping. They also found a new role of HSN, a serotonergic neuron that regulates egg-laying behavior, potentially linking reproduction and feeding behavior. The methods are original and the discoveries are new and exciting.

Suggestions:

1. I understand that the main focus of this paper is the new method and the mode of feeding behavior but I feel that the role of HSN has left little too open-ended: If possible, the mod-5 hypothesis the authors discussed would be one way to address how HSN works. Also does HSN work through ser-1 or ser-4?
The model: Why and how two different sources of 5-HT work distinctly for bursts of the fast pumping? - I apologize if I missed the explanation but the model did not make too much sense especially because the authors assumed that 5-HT worked humorally (and I agree). The 'context dependent' seemed little too vague.

Point-by-point response to reviewers' comments

Reviewer #1

Overall, this is an interesting study. The authors used a custom microfluidic device to deliver precise concentrations of bacterial food to worms while monitoring pharyngeal pumping. The use of this method to monitor pharyngeal pumping is highly innovative and should become the standard in this particular area of research. Increasing food concentration increases the duration and frequency of fast pumping bursts, but each pumping action is of uniform duration regardless of food concentration. The authors use a variety of mutants to demonstrate that serotonin underlies this behavior. The serotonergic neurons NSM and HSN are implicated in for fast pumping behaviors. The receptors SER-1 and SER-4 mediate two different aspects of this behavior.

We thank the reviewer for the encouragement and the thoughtful remarks.

Major:

The major criticism is that the article does not reveal a profoundly novel insight. The authors admit that serotonin is already known to mediate pumping behaviors. The article does contribute many novel findings to the field, including a role for the HSN neurons in regulating pharyngeal behavior and the specific roles of SER-1 and SER-4 receptors in mediating this behavior. The method developed by the authors is highly impressive, and is poised to become the gold standard for this type of research. However, the article would be strengthened in several ways by combining their approach with more sophisticated worm genetics. For instance, the authors use tetanus toxin expression and *twk-18(cn110)* expression in ADF to disrupt ADF signaling and activity, but they don't extend this approach to HSN or NSM neurons to confirm their requirement for pharyngeal pumping. In no case do they kill any of these neurons, either genetically or with laser ablation. This is especially warranted in the case of HSN, which appears to play a novel role in regulating pharyngeal pumping. They suggest the possibility that HSN may be secreting humoral serotonin, which would bypass NSM signaling. Killing NSM in their experiment would test this possibility directly.

As described above, the new version of our paper described experiments in which the NSM and HSN neurons were ablated by reconstituted caspase. The results, which show that NSM alone is required for bursts of fast pumping, indeed distinguish the two possibilities directly, as we describe in the new manuscript (**Figure 4C, Supplementary Figure 4G**).

We also considered other possible techniques. First, unlike in the ADF neurons, no known promoter is expressed exclusively in either the NSM or HSN neurons, preventing targeted expression of the tetanus toxin or *twk-18(cn110)* without a sophisticated genetic construct (that is unlikely to work well). This problem is bypassed by the reconstituted caspase, whose two subunits are expressed from different promoters that intersect exclusively at the target neurons. Second, as noted by the reviewers, our approach is based on quantitative measurements from a robust microfluidic environment. This methodology does not lend itself easily to manipulations that are done one animal at a time, such as laser ablation.

The authors used five null mutants for five serotonin receptors, and one quintuple mutant. While impressive, the use of double mutants would allow for more sophisticated analyses. For instance, the authors cite that *ser-7;ser-4* double mutants have been shown to exhibit behavioral phenotypes not seen in single mutants on agar plates. How would that same double mutant behave in their experimental design? If there are differences between worms on plates and worms in microfluidic plates (the authors highlight several), then direct comparisons of mutant behavior between them are inappropriate.

In the new manuscript we considered two double mutants: *ser-7;ser-4* and *ser-7 ser-1*. The former, which has appeared previously in the literature, has seemed to be lost over the years, despite the best efforts of the Komuniecki lab (which we truly appreciate). We therefore generated this double mutant ourselves. The results are reported in **Figure 5BC, Figure 6DE, and Supplementary Figure 5, 6EF**.

Other comments:

Given that one of major features of this article is their combination of microfluidics with high resolution imaging, it would be good if they could identify the software used for imaging. If it is custom software, then publishing the code would be helpful.

We used ZEN 2012 from Zeiss for image acquisition, and custom software for image processing. Since the latter is rather complex, we offer to share it with any interested researcher upon request (rather than post an impossible-to-follow piece of code online). This information was added in the result section.

In Figure S2, the x-axis label is cut off.

We corrected this unfortunate mistake.

In Figure S2C, the figure indicates the solid lines are the average of the dotted line values. I believe the solid lines are actually the chord lines. If this is the case, then the figure should indicate that.

We see how the legend of this figure could be confusing. We have corrected both the legend and the caption to make it clear that the lines are indeed chord lines.

I was unable to view the movies, so I have no comments on those.

That's unfortunate; we think the movies are informative. We have tried to regenerate the movies, and hope that this solves potential technical problems.

Reviewer #2 (Remarks to the Author):

The manuscript by Lee et al. describes a method monitoring *C. elegans* feeding behavior in liquid media. The authors made use of a microfluidic device, which is composed of an array of chambers with each chamber housing one worm, and performed time-lapse imaging of pharyngeal pumping of individual worms. This group has already published the design and utility of the device. In this manuscript, the authors characterized pumping frequency and duration of individual pumps as a function of bacterial food concentration in the media. Further, the authors tested a series of mutants in neurotransmitter serotonin synthesis and serotonin receptors, and showed that multiple serotonergic neurons and serotonin receptors were involved in food-dependent increase of pumping rates. Although many methods have been used in characterizing *C. elegans* feeding behavior, this method appears easy to use, especially for quantifying detailed pumping events. Likewise, serotonin synthesis and serotonin receptor mutants in food-induced pumping have been reported by several laboratories, but this manuscript identified the involvement of another pair of serotonergic neurons, NSM, in modulating pumping behavior. Overall the work is interesting although it is not entirely new.

We thank the reviewer for the considerate remarks and the creative suggestions. We would like to take the opportunity to clarify an additional important contribution of our work. Previous literature in the field described pumping using a single number, usually termed "rate": ultimately, this is the number of observed pumps over a short observation period, divided by the period duration. Our results indicate that this measure is too crude, and that in fact one should distinguish the number of pumping event from the existence of bursts of fast pumping. This is not a superfluous observation: this distinction allows us to characterize the role of HSN and to discriminate between the roles of the SER-1 and SER-4 receptors.

The following are specific comments to the experiments and data:

1. The role of serotonin in the current feeding assay needs to be clarified. The authors showed that the ADF serotonergic neurons, which have been shown to play a key role in feeding behavior by several laboratories, are not required for the pumping events measured in this assay. In contrast, the authors identified that a pair of HSN serotonergic neurons plays an important role in the current assay. Since HSN neurons do not synthesize serotonin in *C. elegans* larvae, it would be interesting to determine if serotonin regulates feeding behavior only in adult but not in larval worms. This is a simple experiment, should be done.

As described above, we were very excited about this suggestion, despite the fact that this experiment was not as simple as one might think (since larvae really don't like to stay calmly in their chambers). We found that L4 larvae show bursts of fast pumping just like adults, even when the NSM neurons do not synthesize serotonin. In retrospect, this result is very difficult to interpret, and would require a broad array of follow-up experiments, which we plan to perform in the near future.

The authors suggested that one possibility is that NSM neurons uptake serotonin released from HSN to in turn regulate feeding. This possibility is unlikely, as laser ablation of NSM did not reduce pumping rates (Avery & Horvitz, Neuron, 1989). In fact, biochemical pathways and neural circuits involved in *C. elegans* feeding behavior have been studied extensively and many mutants are available. Some experiments to place serotonin and serotonergic neurons into the behavioral circuits could offer new mechanistic insights into serotonin regulation of feeding behavior, and enhance general

interests to the paper.

Indeed, this is a somewhat provocative suggestion, which we also thought was unlikely given the Avery & Horvitz paper (which was cited in our paper). But the only remark made in this paper about the NSM laser ablation behavior was that it showed “normal pumping”. We therefore generated new transgenic lines that express reconstituted caspase to ablate the HSN and NSM neurons, independently. These results show that while NSM ablated neurons are capable of fast pumping, they do not show the bursts of fast pumping that characterize feeding by wild-type animals, and therefore show considerable reduction in the pumping rate. This suggests that the NSM neurons are indeed required for regulating pumping, even when serotonin is synthesized only in the HSN neurons.

We hypothesized that the reuptake receptors MOD-5 may be involved in the process in which NSM uses HSN-synthesized serotonin. As a quick and crude attempt at testing this hypothesis, we therefore characterized pumping in *mod-5* mutants. However, these worms display consistent fast pumping that clouds the observation of any quantitative phenotype, presumably because lack of MOD-5 causes endogenous serotonin to persist in synapses after release.

2. Experiments over serotonin receptor mutants need to be improved to make the points clear. Figure 5B showed that serotonin synthesis mutant *tph-1* was defective in fast pumping, but deletion mutants of any of the five identified C. elegans serotonin receptors or quintuplet mutants of all the serotonin receptors largely preserved fast pumping frequency, suggesting that serotonin can stimulate pumping in the absence of these receptors. However, Figure 6E showed that the quintuplet serotonin receptor mutants completely failed to respond to exogenous serotonin.

We too are fascinated by these results, which display genetic interactions among 5-HT receptors that presumably reflect the complexity of the underlying signaling network. Characterizing this network is a major endeavor, which we plan to pursue in the immediate future. With too many combinations to be explored, we focused here on combinations of the receptors we previously identified (*ser-7 ser-1* and *ser-7;ser-4*).

That the quintuple mutant does not respond to exogenous 5-HT is presumably the effect of *ser-7* (indeed the two double mutants we added to the work, *ser-7 ser-1* and *ser-7;ser-4* show the same behavior).

One caveat is that the data were entirely rested on single mutant allele of the receptors. The authors should validate the results by testing multiple available serotonin receptor alleles, and performing transgenic rescue of the phenotype.

The alleles used in this study are all widely used in the field. Surprisingly, no other alleles are available for any of the serotonin receptors we characterized (*ser-1*, *ser-4*, and *ser-7*). Since the phenotypes we described for the latter have been described in the past, we focused on the former two, where we identified new phenotypes, and generated new null alleles using CRISPR-Cas9. Our results are described above and in **Figure 5BC, Figure 6DE, Supplementary Figure 5, 6EF**.

It also turns out that toxic effects make transgenic rescue surprisingly difficult, as confirmed by experts in the field. We were able to establish a stable rescue line for *ser-1*, but not for *ser-4*. The results are described in **Supplementary Figure 5E**.

In addition, to determine gene function in behavior, typically sibling worms of each genotype with or without environmental stimuli or drug treatments should be assayed in parallel, thus the basal pumping rates and the ability to respond to food and drugs be compared.

In our assay, worms always spend more than two hours in the device under standard conditions. We switch the environment to the experimental conditions (with different amounts of food, drugs etc) only after establishing base-line pumping dynamics. Thus, the microfluidic approach provides a way to establish a better standard than the sibling-based methodology one has to use in plate-based assays. We described this in the revised materials and methods.

3. The authors should provide more details about the experiments: how many animals were tested for each experiment, how many independent trials were performed for each assay, and which differences were statistically significant.

We regret this omission. These details are now provided in the revised materials and methods, Supplementary Table 2, and statistical significance is indicated in the figures.

Reviewer #3 (Remarks to the Author):

Summary:

Lee et al developed a new method to measure how feeding motion is regulated by food availability and concentrations in *C. elegans* using microfluidic device and long-term imaging tools. From the study, they discovered that *C. elegans* regulates food intake by altering the frequency of bursts of fast pumping. They also found a new role of HSN, a serotonergic neuron that regulates egg-laying behavior, potentially linking reproduction and feeding behavior. The methods are original and the discoveries are new and exciting.

We truly appreciate the reviewer's mindful comments and encouragement.

Suggestions:

I understand that the main focus of this paper is the new method and the mode of feeding behavior but I feel that the role of HSN has left little too open-ended: If possible, the mod-5 hypothesis the authors discussed would be one way to address how HSN works. Also does HSN work through ser-1 or ser-4?

As described above, we addressed the role of the HSN neurons by ablating the NSM and HSN neurons, and found that HSN alone is neither sufficient nor required for pumping, suggesting that HSN-synthesized serotonin still requires NSM.

Testing the mod-5 hypothesis requires some non-trivial genetics, in which mod-5 would be deleted only in the NSM neurons, as this gene is widely expressed. When we characterized pumping in mod-5 mutants, we found that these worms display consistent fast pumping that clouds the observation of any quantitative phenotype, presumably because lack of MOD-5 causes endogenous serotonin to persist in synapses after release.

The model: Why and how two different sources of 5-HT work distinctly for bursts of the fast pumping? - I apologize if I missed the explanation but the model did not make too much sense especially because the authors assumed that 5-HT worked humorally (and I agree). The 'context dependent' seemed little too vague.

With the new results from ablation experiments, we revised our model.

Life in the microfluidic device is different in many ways from life on an agar plate (let alone life in a rotting fruit). For example, on a plate worms go in and out of the bacterial lawn, switching back and forth between "OD=0" and "OD>6". In a previous work we (and others) have shown that the ADF neurons are involved in the transition between these two environments. It is possible that under such conditions, ADF can also be involved in facilitating pumping. This hypothesis, for example, is what we meant by "context dependent". In the new manuscript we try to be somewhat more explicit, without becoming too speculative.

The question why endogenous and exogenous remains unanswered. Supply of exogenous serotonin is a very unnatural situation, in which 5-HT enters the animal presumably through the intestine. The identity of the synapses it ends up at, the local concentration there, or its temporal profile are all unknown, and there is no reason to think it would mimic endogenous serotonin in any of these and other aspects. We (and others) showed that the SER-7 receptors are required for induction of pumping by exogenous 5-HT, but not by food.

References

1. Desai, C., Garriga, G., McIntire, S.L. & Horvitz, H.R. A genetic pathway for the development of the *Caenorhabditis elegans* HSN motor neurons. *Nature* **336**, 638-46 (1988).
2. Chelur, D.S. & Chalfie, M. Targeted cell killing by reconstituted caspases. *Proceedings of the National Academy of Sciences* **104**, 2283-2288 (2007).
3. Avery, L. & Horvitz, H.R. Pharyngeal pumping continues after laser killing of the pharyngeal nervous system of *C. elegans*. *Neuron* **3**, 473-85 (1989).
4. Hobson, R.J. et al. SER-7, a *Caenorhabditis elegans* 5-HT7-like receptor, is essential for the 5-HT stimulation of pharyngeal pumping and egg laying. *Genetics* **172**, 159-69 (2006).

5. Sze, J.Y., Victor, M., Loer, C., Shi, Y. & Ruvkun, G. Food and metabolic signalling defects in a *Caenorhabditis elegans* serotonin-synthesis mutant. *Nature* **403**, 560-4 (2000).
6. Kullyev, A. et al. A genetic survey of fluoxetine action on synaptic transmission in *Caenorhabditis elegans*. *Genetics* **186**, 929-41 (2010).

REVIEWERS' COMMENTS:

Reviewer #1 (Remarks to the Author):

The authors have addressed my comments.

Reviewer #2 (Remarks to the Author):

In the revised manuscript, the authors clarified and strengthened the potential utility of the method for characterizing *C. elegans* feeding behavior. However, even with the new experimental data, the mechanisms of serotonergic neurons, serotonin receptors, or serotonin in regulating *C. elegans* feeding behavior in the current experimental paradigm remain unclear. Also, some statements in the manuscript are not supported by data and confusion. For instance, "regulation of bursts is a conserved mechanism of behavior and motor control". In fact, this manuscript could be improved if the authors can clearly outline a few sentences what is the new mechanistic insight revealed from their studies, and how their findings could be interesting to broad readers of Nature Communications.

Reviewer #3 (Remarks to the Author):

The authors did a great job to answer all my comments. I do not have any further suggestions.

Point-by-point response to reviewers' comment

Reviewer #2

In the revised manuscript, the authors clarified and strengthened the potential utility of the method for characterizing *C. elegans* feeding behavior. However, even with the new experimental data, the mechanisms of serotonergic neurons, serotonin receptors, or serotonin in regulating *C. elegans* feeding behavior in the current experimental paradigm remain unclear. Also, some statements in the manuscript are not supported by data and confusion. For instance, "regulation of bursts is a conserved mechanism of behavior and motor control". In fact, this manuscript could be improved if the authors can clearly outline a few sentences what is the new mechanistic insight revealed from their studies, and how their findings could be interesting to broad readers of *Nature Communications*.

We find the suggestion to spell out the new insights revealed in this study very useful. We added a paragraph towards the end of the discussion section that clarifies these findings and their implications.

The statement "regulation of bursts is a conserved mechanism of behavior and motor control" that appeared in the previous version of the manuscript was not meant to suggest an evolutionary conservation, which would require data we do not have, but simply that such mechanism is abundant in nature. We have rephrased this statement in the last paragraph of the text.